# The speed of sight: Individual variation in critical flicker fusion thresholds

**Clinton S. Haarlem**[1,2]*, **Redmond G. O'Connell**[2], **Kevin J. Mitchell**[2,3], **Andrew L. Jackson**[1]

**1** Department of Zoology, Trinity College Dublin, Dublin, Ireland, **2** Trinity College Institute of Neuroscience, Trinity College Dublin, Dublin, Ireland, **3** Smurfit Institute of Genetics, Trinity College Dublin, Dublin, Ireland

* haarlemc@tcd.ie

**Data Availability Statement:** All relevant data can be accessed at Zenodo repository (Link: https://doi.org/10.5281/zenodo.10694589)

**Funding:** Funding was provided by Trinity College Dublin under the Provost's Postgraduate Award

## Abstract

The critical flicker fusion threshold is a psychophysical measure commonly used to quantify visual temporal resolution; the fastest rate at which a visual system can discriminate visual signals. Critical flicker fusion thresholds vary substantially among species, reflecting different ecological niches and demands. However, it is unclear how much variation exists in flicker fusion thresholds between healthy individuals of the same species, or how stable this attribute is over time within individuals. In this study, we assessed both inter- and intra-individual variation in critical flicker fusion thresholds in a cohort of healthy human participants within a specific age range, using two common psychophysical methods and three different measurements during each session. The resulting thresholds for each method were highly correlated. We found a between-participant maximum difference of roughly 30 Hz in flicker fusion thresholds and we estimated a 95% prediction interval of 21 Hz. We used random-effects models to compare between- and within-participant variance and found that approximately 80% of variance was due to between-individual differences, and about 10% of the variance originated from within-individual differences over three sessions. Within-individual thresholds did not differ significantly between the three sessions in males, but did in females (P<0.001 for two methods and P<0.05 for one method), indicating that critical flicker fusion thresholds may be more variable in females than in males.

## Introduction

Visual perception is one of the most salient types of sensory processing found in the animal kingdom, with around 96% of known animal species alive today possessing image forming eyes [1]. Visual systems in nature are incredibly diverse and their specific functions and sensitivities vary greatly depending on a species' ecological demands. However, all sensory processing is constrained by the maximum rate at which the system can parse information. This processing rate is different among species and is known as their temporal resolution [2–4]. A finely tuned visual temporal resolution is necessary to respond appropriately to a dynamic environment and it determines a species' ability to gauge how visual scenes change over time and how accurately it can detect rapid changes in light intensity. The processing of both spatial

2019 scheme to ALJ, KJM and RO'C. The funders had no role in study design, data collection and analysis, decision to publish, or preparation of the manuscript.

**Competing interests:** The authors have declared that no competing interests exist.

and temporal information is also crucial for motion perception, and the resolution with which this occurs may offer distinct benefits and drawbacks. A high temporal resolution may offer an increased ability to track fast-moving objects in high luminance conditions, whereas a lower temporal resolution might be more advantageous for the extrapolation of the direction of motion of slower objects and for increased temporal integration under low luminance conditions [5,6].

One commonly used method to quantify visual temporal resolution is with the critical flicker fusion (CFF) threshold: the point at which an intermittently flashing light flashes at such high a frequency, that individual flashes can no longer be discriminated and the light appears constant. An individual's CFF threshold is dependent on the physical properties of the stimulus, such as its intensity, wavelength, size and its distance from the observer [7–9]. As all visual systems rely on the integration and summation of luminous energy, CFF is applicable and generalizable across species, making it a useful proxy for visual temporal resolution research. Using simple experimental paradigms, CFF thresholds can also be determined in several ways: either behaviourally with trained animals, or electro-physiologically with electroretinograms.

As variation between individuals of the same species is present in many aspects of visual perception, we aimed to evaluate whether this is also the case for critical flicker fusion thresholds, using healthy human participants. CFF has been well studied in humans, and previous research has found that critical flicker fusion thresholds are affected by many physiological factors [10]. For instance, different parts of our retina have different levels of sensitivity to flicker [11,12]. Since flickering visual stimuli are processed by the nervous system, CFF thresholds may also be affected by impaired cerebral functioning [13,14] and by neurological conditions such as Alzheimer's disease [15].

Certain physiological stresses such as exercise or physical exhaustion may directly, but temporarily, affect an individual's CFF threshold [16–18]. It has also been suggested that CFF might be used as a measure for cortical arousal and central nervous system activity [19–22]. Flicker fusion thresholds may even be affected by an individual's blood oxygen content [23]. CFF thresholds also tend to be lower in older people than in younger people [24–26].

Due to numerous external and internal factors that may affect CFF measurements, many different methods and techniques have been designed to quantify this trait, making it extremely difficult, if not impossible, to compare study results directly. Some methods, such as the classical psychophysical "method of constant stimuli," require a large number of repetitions of each stimulus to produce an accurate measure of performance, resulting in extremely long testing times and subsequently introducing fatigue as a possible confounding factor [27,28].

Additionally, CFF studies often rely on small sample sizes and thresholds are generally only reported as group averages, for example when assessing the effects of sex or age on CFF thresholds [25,29,30]. Previous research has also placed considerable focus on how CFF thresholds may be affected by physiological or psychological phenomena, or by pharmacological intervention [31–34]. This research is valuable, but leaves the question of how much natural variation exists in CFF thresholds, and whether varying thresholds represent stable individual traits or more fluctuating variation due to other factors.

In the current study, we hypothesized that a considerable amount of natural individual variation exists in CFF thresholds. Additionally, we hypothesized that CFF thresholds are a stable trait within individuals. Quantitatively these predictions will manifest as larger variation among individuals compared to the variation within individuals. To quantify this, we employed a modified combination of several commonly used testing methods to address several questions:

1. We measured the amount of natural variation in CFF thresholds in a healthy cohort of similar age, in the absence of other stressors.

2. We measured the test-retest stability of CFF thresholds over time in order to quantify within-individual variation.

3. We assessed the level of correlation between the different testing methods, and the robustness of using a testing protocol that drastically decreases testing time by combining several methods.

## Methods

### Participants

88 participants (53 female, age range 18–35 years) were recruited from Trinity College Dublin. 47 participants completed the task as part of a larger perception study. 11 of these participants were researchers in the School of Psychology and participated voluntarily. The remaining 36 were compensated with either a 10 Euro gift voucher per hour or by teaching credits forming part of their postgraduate research degree at the rate of 1 credit per half-hour of participation, according to the European Credit Transfer System (ECTS). 41 participants completed only the CFF threshold task associated with this study and participated on a voluntary basis. All participants provided written, informed consent and all participants had normal or corrected-to-normal visual acuity. The study was approved by the Ethics Committee of Trinity College Dublin's School of Psychology (approval ID SPREC052022-01). Participant recruitment and data collection commenced on 1 August 2022 and ended on 6 July 2023.

### Apparatus

We measured critical flicker fusion thresholds using a novel measuring device, consisting of an Arduino Uno R3 microprocessor and a standard 5mm diameter, round-lens 4500K white LED light, housed in a black container. An opaque, black viewing tube and goggles with blacked out frames were mounted on top of the container, to standardize viewing distance at ~16 cm and to eliminate almost all ambient light. Flicker frequency of the LED could be adjusted with a rotary encoder dial that either increased or decreased flicker rate in 1 Hz increments (Fig 1). We measured luminance and colour temperature with a ColorCal Mk II. Mean luminance of the LED was ~255 lux. The device was powered by a personal laptop, using a USBC cable. We measured the power input with a Tektronix TDS 210 oscilloscope to ensure stability. The precision and stability of the LED flash timings were measured using a photometer, both before the start of, and after completion of the study. Participants completed the task while seated, with the box placed in front of them on a desk. Participants were allowed to hold or tilt the box as needed to suit their comfort and to prevent neck- or back strain while operating the device.

### Testing protocol

The experiment was conducted throughout the day, between 9am and 5:30pm. The protocol consisted of a combination of two commonly used psychophysical experimental techniques: the method of limits (MOL) and the method of constant stimuli (MOC). As the method of limits generally produces a fast result, but the method of constant stimuli has a higher level of precision and stability [27], we combined the two methods to produce a protocol that took less than 10 minutes to complete. The MOL can be employed in two distinct manners (ascending

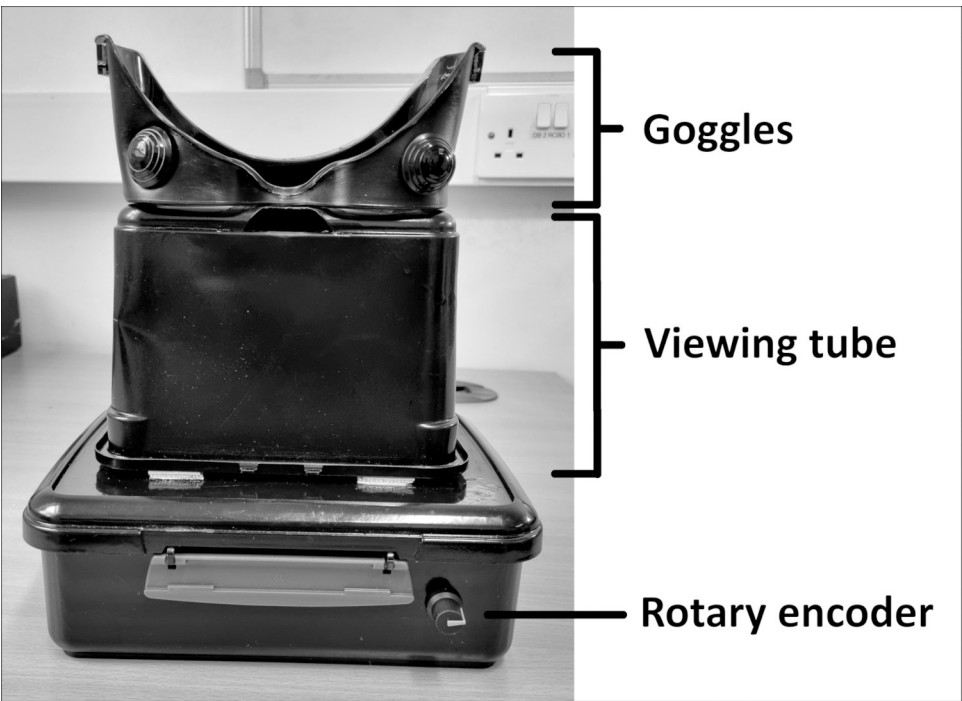

**Fig 1. CFF measuring apparatus.** The LED light and electronic components were housed in an opaque, black box. On top were mounted a viewing tube and goggles with blacked-out frame.

and descending) and we used both, to give us three different CFF threshold values per participant, per session.

The protocol started with two different measurements of the method of limits: participants observed a constantly lit LED light through the viewing tube and were then instructed to turn the rotary encoder clockwise, to make the light source start flashing. Participants were then asked to keep rotating the dial to increase the flash frequency in 1 Hz increments until flashes could no longer be perceived and the light appeared steady. The frequency at which this occurred was recorded (ascending measurement). The LED light source was then set to a flash frequency of 65 Hz, which is above the threshold at which humans are generally able to perceive light flashes at the described intensity. For the second measurement, participants were instructed to observe the light source again, but this time they were asked to turn the dial counter-clockwise to decrease flash frequency, until they first observed flashing. The frequency at which this happened was again recorded (descending measurement). One participant indicated during one session that they were able to perceive flashing at the starting point of 65 Hz. For this session, the starting point was set to 80 Hz instead.

The third step consisted of the method of constant stimuli: for each participant, we took the highest of the two aforementioned recorded thresholds, and created a series of 10 flash frequencies in 1 Hertz steps: 5 frequencies below the recorded threshold, and 4 frequencies above it. Each of the frequencies in the series was then multiplied by 5 to produce a set of 50 stimuli. We then randomized the entire set and presented it to the participant. Participants were then instructed to cycle through all of the 50 frequencies in the series, one by one, using the rotary encoder dial to instantiate the next stimulus in the set. Participants were asked to inform the experimenter for each stimulus whether they observed a steady light or not. The final critical flicker fusion threshold for the individual was determined by calculating the highest frequency in the series at which the user was able to perceive flashing 80% of the time.

## Within-individual variation

To investigate the stability of CFF thresholds within individuals, 49 participants completed the task three times, with a minimum of 24 hours between each measurement. Each participant was retested during the same part of the day: either in the morning or in the afternoon.

## Statistical analysis

We performed statistical analyses using R version 4.2.1 [35] and RStudio version 2023.3.0.386 [36].

We calculated method of constant stimuli thresholds by creating a binomial generalized linear model for each participant, using the proportion of responses per frequency at which flicker was perceived as response variable and the frequency in Hertz as the explanatory variable. In these models, the range for $y$ was set to 0–1, with 0.8 corresponding to an 80% correct score. We therefore calculated the x-value at which y = 0.8 (Fig 2).

We analysed the between- and within- individual variation with random mixed effects models, using the lme4 package [37] and we used Pearson's correlation coefficient to analyse the level of correlation between the three different measurements.

Normality of the data was assessed numerically with Shapiro tests and visually with quantile-quantile-plots of model residuals. Quantile-quantile plots were created using the car package, which contains functions and tools for statistical regression analysis [38].

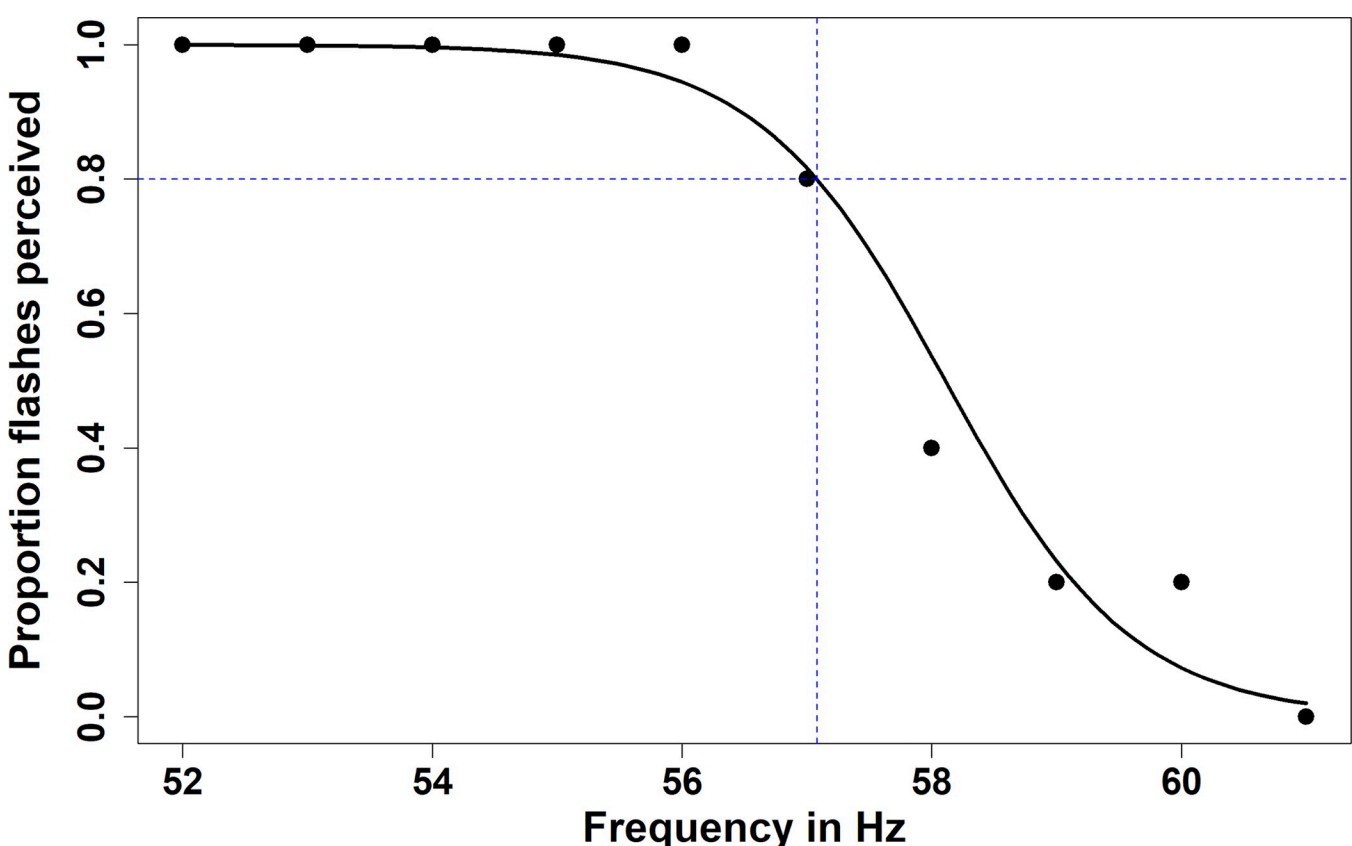

**Fig 2. Example of CFF-threshold calculation for the method of constant stimuli.** We created and plotted binomials models for each participant. The CFF threshold was then calculated for each participant as the x-coordinate at which the proportion of correct responses was equal to 0.8. Shown here as the intersection point of the two dashed blue lines, where y = 0.8 and x = 57.08.

CFF thresholds followed a normal distribution for the method of constant stimuli (P = 0.12) and descending method of limits (P = 0.57), but not for the ascending method of limits (P < 0.05). Visual inspection indicated this was likely due to two extremely low values of 20 and 21 Hz. (Fig 2.) Quantile-quantile plot indicated a normal distribution otherwise, and Shapiro-Wilks test with the two outliers excluded confirmed normality of the data (P = 0.07). As the two outliers represent natural population variation, they were not excluded from the dataset and parametric analyses were carried out for all three methods.

## Results

### Correlation between methods

We obtained three CFF-threshold values for each participant (n = 88): one using the ascending method of limits, one using the descending method of limits and one using a shortened version of the method of constant stimuli. We found a high level of correlation (P<0.001) between all three methods, with Pearson's $r = 0.93$ between MOC and MOL descending, $r = 0.76$ between MOC and MOL ascending and $r = 0.68$ between MOL descending and MOL ascending (Fig 3).

### Variation in CFF thresholds

We tested a total of 88 participants and recorded three CFF threshold measurements for each participant. The distribution of CFF values was similar for each method (Fig 4). Table 1 lists the results for each method used.

Results of CFF measurement for the method of constant stimuli (MOC), ascending method of limits (Asc. MOL) and descending method of limits (Desc. MOL). Columns indicate method, minimum and maximum measured values, group mean, standard deviation (SD), variance and 95% prediction interval (PI), respectively.

We found no difference in mean CFF-thresholds between males and females for any of the methods (P>0.05) (Fig 5).

### Test-retest variability

Of the 88 participants, 49 participants completed three sessions of CFF measurements. For each method, all three sessions were highly correlated with each other (Table 2). Within- and

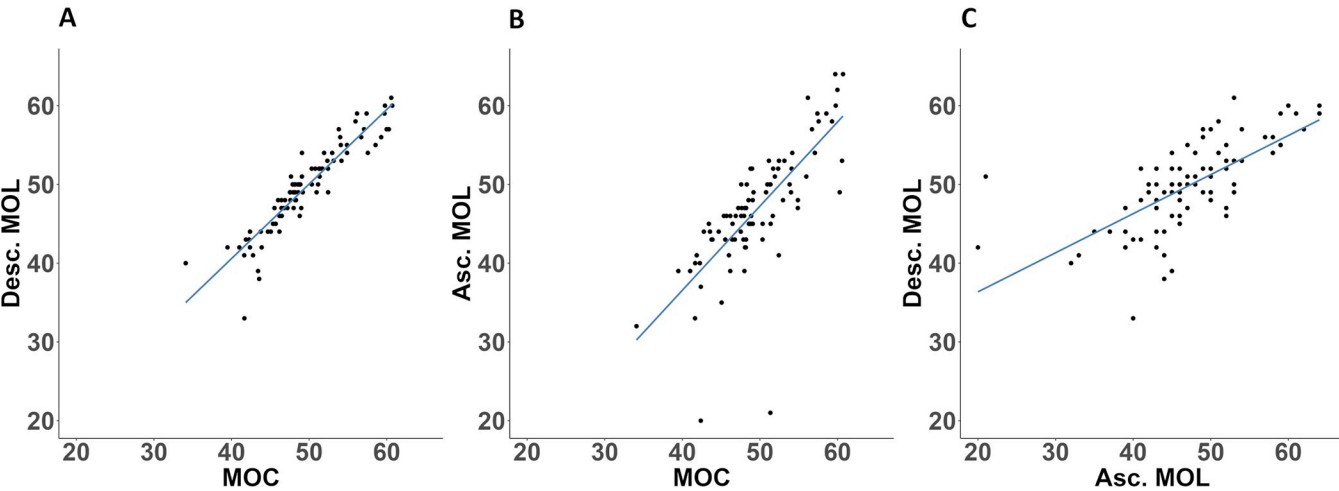

**Fig 3. Correlation between the different methods.** A: Correlation between MOC and MOL descending, b: Correlation between MOC and MOL ascending, and c: Correlation between MOL ascending and MOL descending.

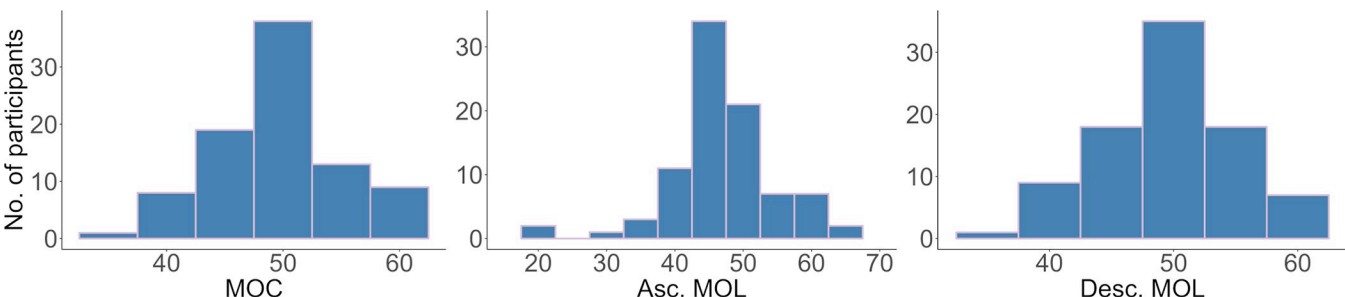

**Fig 4. Distribution of CFF thresholds for the three different measurements.** The method of constant stimuli and descending method of limits both followed a normal distribution (Shapiro-Wilk normality test: P = 0.12 and P = 0.57 respectively). Due to two influential extremely low values, the ascending method of limits does not follow a normal distribution (P < 0.05).

between individual variation in CFF thresholds for all three methods were analysed using mixed effects models, where the CFF threshold was used as the response variable, session as explanatory variable, sex as a fixed effect and participant as random effect. As we hypothesized that the magnitude of within-participant variation might not be equal between participants, we performed the analyses using both random intercept and random slope models. The random slopes allow for different rates of change in CFF over the course of the three sessions. We used ANOVA to compare full (random slope and intercept) models against random-intercept-only models and against null models. To investigate possible effects of sex on CFF, best-fitting models were also compared against similar models that did not include the "sex" variable.

For all three methods, data was best explained with the inclusion of a random intercept and slope. For all three methods, full models were significantly different from random-intercept-only models and null-models, indicating that CFF thresholds within the same participants differed between sessions (Table 2. P<0.01 for all methods). Session correlated positively with CFF for all three methods, with 0.89 Hz for MOC, 1.22 Hz for ascending MOL and 1.3 for descending MOL, which accounted for about 10% of the variance for each method. About 80% of the variance in the multiple-measures data was explained by between-participant differences. The remaining 4–9% variance can be assumed to be due to within-individual differences (Table 3).

## Sex differences in within-individual variation

For the descending MOL, the model including sex was significantly different from the model where it was not included as a fixed variable (P<0.05) and for the MOC the two models were near-significantly different (P = 0.055). For both models AIC values were lower with sex included as a fixed variable. For the ascending MOL, AIC values were similar and removing the variable "sex" did not have an effect on the model (P = 0.29). Due to this possible effect of sex on the models, we conducted a post-hoc analysis of the multiple-measures data for each sex separately. We again created random slope fixed effects models with CFF threshold for

**Table 1. Comparison of the three CFF measuring methods.**

| Method | Min. | Max. | Mean | SD | Variance | 95% PI |
|---|---|---|---|---|---|---|
| MOC | 34.09 | 60.71 | 49.59 | 5.41 | 29.29 | ± 10.6 |
| Asc. MOL | 20 | 64 | 46.81 | 7.63 | 58.27 | ± 15 |
| Desc. MOL | 33 | 61 | 49.66 | 5.54 | 30.69 | ± 10.9 |

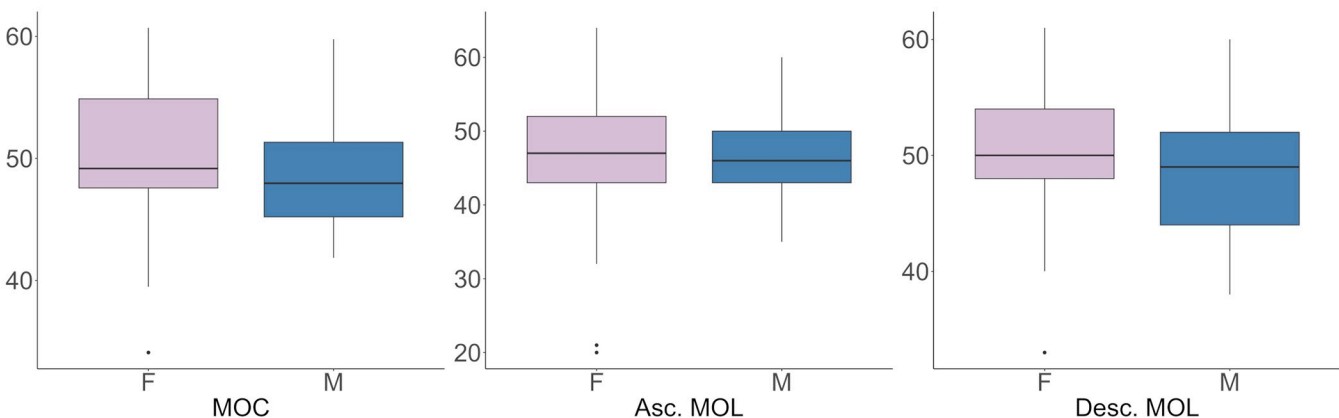

**Fig 5. Boxplots of CFF-thresholds for each method, separated by sex.** T = 1.93 for MOC, t = 0.54 for Asc. MOL and t = 1.96 for Desc. MOL. M = males, F = females.

each method as the response variable, session as explanatory variable, and participant as a random effect. This time, for all three methods, full models were no different from null-models for male participants (n = 20), indicating that for males, CFF-thresholds were relatively stable between sessions (P = 0.38 for MOC, P = 0.17 for asc. MOL and P = 0.31 for desc. MOL). For females (n = 29) however, full models were significantly different from null models for each method (P<0.001 for MOC and desc. MOL, and P<0.05 for asc. MOL) (Fig 6). In males, CFF increased by about 0.4 Hz. between sessions on average, whereas in females it increased by about 1.6 Hz (Table 4).

## Discussion

With our current study, we have validated a quick, low-cost and portable method to test CFF thresholds. Results from all three methods followed a similar range and standard deviation (Table 1). The CFF threshold range discovered for the method of constant stimuli is also in agreement with the one found in Eisen-Enosh et al. [27]. Similar to our results, Eisen-Enosh et al. also reported a slightly lower test-retest repeatability for the method of limits than for the method of constant stimuli. This, according to our findings, may in most part be due to the ascending part of the measurement. Our results indicate that a shortened version of the method of constant stimuli can still provide an accurate result, while greatly reducing testing time. The ascending method of limits may be slightly more prone to extreme results. It is common in psychophysical paradigms to calculate a mean threshold from only an ascending- and a descending method of limits measurement [27,29,39], which may skew the results and may not represent an individual's true CFF threshold if one of the values turns out to be an outlier.

**Table 2. Correlations between each session for each method.**

| Method | S1 –S2 | | S1 –S3 | | S2 –S3 | |
|---|---|---|---|---|---|---|
| | R | P | R | P | R | P |
| MOC | 0.9 | <0.001 | 0.82 | <0.001 | 0.9 | <0.001 |
| Asc. MOL | 0.61 | <0.001 | 0.43 | <0.01 | 0.82 | <0.001 |
| Desc. MOL | 0.83 | <0.001 | 0.72 | <0.001 | 0.86 | <0.001 |

The level of correlation is indicated with Pearson's *r*, between session one and two (S1-S2), session one and three (S1-S3) and session two and three (S2-S3).

**Table 3. Overall mean CFF, mean CFF difference between sexes and variance distribution for each method.**

| Method | Mean CFF | Female–male difference in Hz | Between-participant variance | Session variance | Residual variance |
|---|---|---|---|---|---|
| MOC | 49.88 | -2.56 | 17.20 | 1.88 | 1.8 |
| Asc. MOL | 46.32 | -1.58 | 56.91 | 8.43 | 6.36 |
| Desc. MOL | 49.76 | -3.03 | 26.73 | 3.37 | 3.17 |

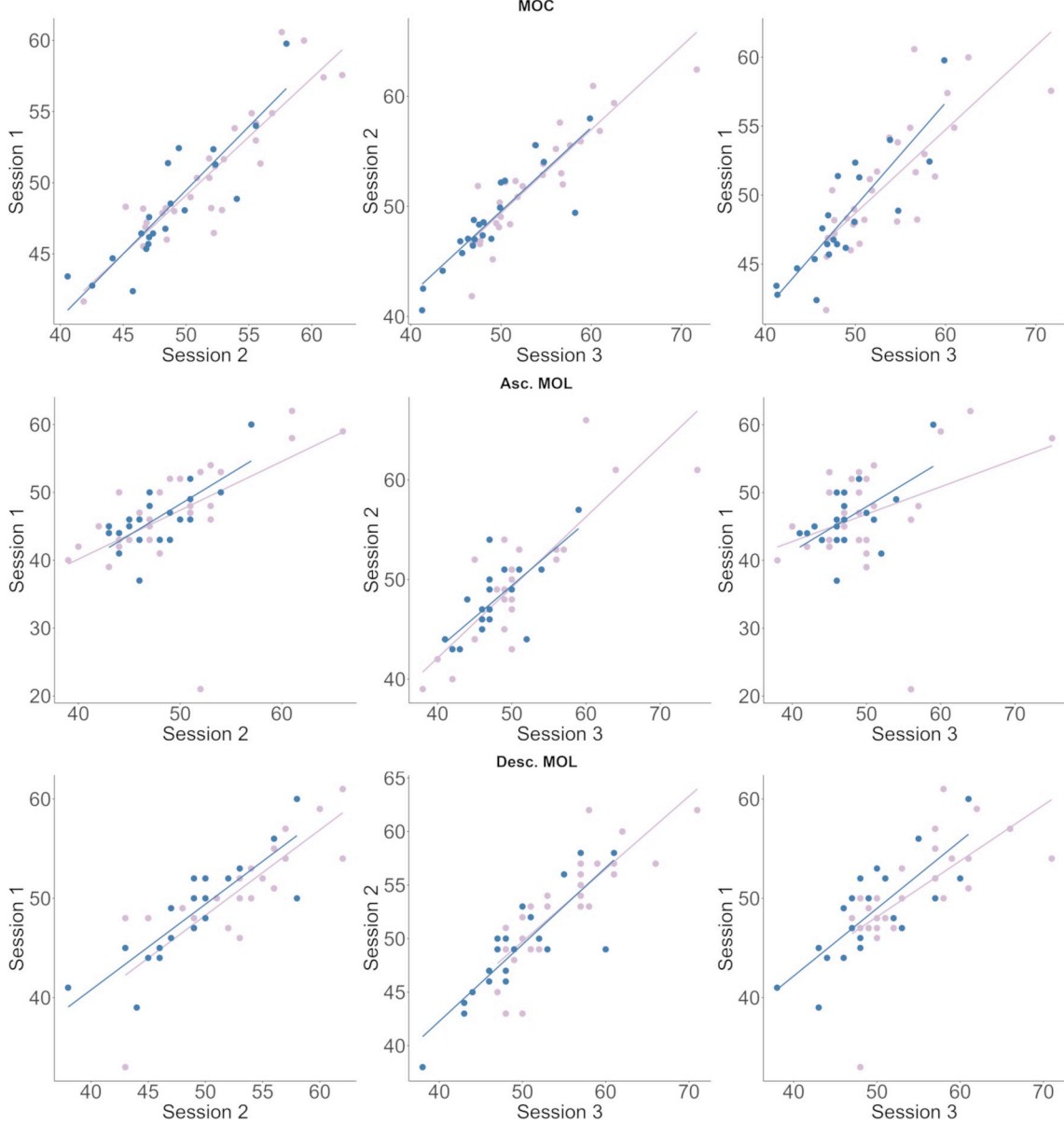

**Fig 6. CFF test-retest variability for males and females.** Correlation plots comparing CFF-thresholds between session one and two, session two and three and session one and three for the method of constant stimuli (top row), the ascending method of limits (middle row) and the descending method of limits (bottom row). Male CFF displayed in blue, female CFF displayed in purple.

**Table 4. Model summaries of multiple-measures CFF per method for each sex.**

| Method | Mean CFF | Difference between sessions in Hz | Between-participant variance | Session variance | Residual variance |
|---|---|---|---|---|---|
| Male MOC | 48.02 | 0.24 | 13.55 | 0.45 | 2.08 |
| Male asc. MOL | 45.93 | 0.63 | 22.67 | 1.78 | 4.23 |
| Male desc. MOL | 48.12 | 0.4 | 20.06 | 1.39 | 3.35 |
| Female MOC | 49.4 | 1.34 | 19.16 | 2.37 | 1.6 |
| Female asc. MOL | 45.49 | 1.64 | 78.86 | 12.6 | 7.83 |
| Female desc. MOL | 48.81 | 1.91 | 29.08 | 3.79 | 3.04 |

CFF thresholds may potentially be made more robust by adding the shortened version of the method of constant stimuli.

Much of the existing literature relating to CFF compare measured thresholds to other physical and/or psychophysical attributes. Our study placed focus on variation in CFF thresholds both within- and between individuals of similar age. With this study, we have shown that there are considerable individual differences in CFF thresholds and that CFF is relatively stable over time. We found no significant difference in mean CFF thresholds between males and females, which is in agreement with previous findings [25,26,40,41]. However, the results from our post-hoc analysis do show a clear difference between sexes in the test-retest repeatability of CFF. This could be an indication that CFF might be more stable in males than in females, but replication studies are needed to further explore this possibility.

As all participants in our study were within a small age range, had normal or corrected-to-normal visual acuity and all declared to be in general good health, the observed variation in CFF thresholds cannot be explained by variation in these factors. However, visual perception involves many physiological and psychological processes. It is therefore difficult to determine from where this variation might arise. For example, at the physical level, differences in the anatomy of the primary visual cortex can affect the processing of visual information [42]. At the higher level of psychological processing, the formation of a percept and subsequently, the generation of a decision about that percept, also influence whether an individual determines they are perceiving a stimulus as constant or not-constant. Additionally, natural variation in brain wave oscillations may result in a difference in visual integration window sizes in different individuals [43].

The magnitude of the variation in CFF we found between individuals is quite large, and we calculated a prediction interval of roughly 21 Hz (± 10.6 Hz from the calculated mean). This amount of variation is comparable with the variation reported among closely related species that occupy different ecological niches, such as having habitats with different luminance levels [44,45] or having a preference for faster or slower moving prey [46]. This suggests that the spread of CFF thresholds in humans is in a range that could have important effects on function. It would therefore be interesting to explore how individual variation in visual temporal resolution in humans may translate to perception and behaviour in real-world scenarios, especially those requiring high-speed perception and action, such as athletic performance or competitive gaming.

## Acknowledgments

We would like to thank all volunteers who participated in this study.

## Author Contributions

**Conceptualization:** Clinton S. Haarlem, Redmond G. O'Connell, Kevin J. Mitchell, Andrew L. Jackson.

**Data curation:** Clinton S. Haarlem.

**Formal analysis:** Clinton S. Haarlem.

**Funding acquisition:** Redmond G. O'Connell, Kevin J. Mitchell, Andrew L. Jackson.

**Investigation:** Clinton S. Haarlem.

**Methodology:** Clinton S. Haarlem, Redmond G. O'Connell, Kevin J. Mitchell, Andrew L. Jackson.

**Project administration:** Clinton S. Haarlem.

**Resources:** Redmond G. O'Connell, Kevin J. Mitchell, Andrew L. Jackson.

**Software:** Clinton S. Haarlem, Andrew L. Jackson.

**Supervision:** Redmond G. O'Connell, Kevin J. Mitchell.

**Validation:** Clinton S. Haarlem, Redmond G. O'Connell, Andrew L. Jackson.

**Visualization:** Clinton S. Haarlem.

**Writing – original draft:** Clinton S. Haarlem.

**Writing – review & editing:** Clinton S. Haarlem, Redmond G. O'Connell, Kevin J. Mitchell, Andrew L. Jackson.

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
