## [Decision Letter · Decision Letter 0]

9 Jan 2024

PONE-D-23-39116The speed of sight: individual variation in critical flicker fusion thresholdsPLOS ONE

Dear Dr. Haarlem,

Thank you for submitting your manuscript to PLOS ONE. After careful consideration, we feel that it has merit but does not fully meet PLOS ONE’s publication criteria as it currently stands. Therefore, we invite you to submit a revised version of the manuscript that addresses the points raised during the review process.

The manuscript has received two reviews. Both suggested minor, but important, revisions that I think will improve the manuscript and should not be burdensome for the authors.

We look forward to receiving your revised manuscript.

Kind regards,

Christopher Nice, Ph.D.

Academic Editor

PLOS ONE

Journal Requirements:

Reviewers' comments:

Reviewer's Responses to Questions

**Comments to the Author**

1. Is the manuscript technically sound, and do the data support the conclusions?

Reviewer #1: Yes

Reviewer #2: Yes

2. Has the statistical analysis been performed appropriately and rigorously? 

Reviewer #1: Yes

Reviewer #2: Yes

3. Have the authors made all data underlying the findings in their manuscript fully available?

Reviewer #1: Yes

Reviewer #2: Yes

4. Is the manuscript presented in an intelligible fashion and written in standard English?

Reviewer #1: Yes

Reviewer #2: Yes

5. Review Comments to the Author

Reviewer #1: This manuscript provides a succinct and skilfully crafted analysis of individual variations in critical flicker fusion frequency thresholds. Three distinct protocols were executed, and the ensuing results are meticu-lously presented, accompanied by judicious statistical analysis.

Below, you will find some minor comments for your consideration.

Minor comments:

l. 59: may be mention: Muth T et al, 2023

l. 99: ECTS, meaning what?

l. 105: a photograph or a schematic with detailed description would be helpful for the less informed reader. Please, add size of the light source.

l. 115: Protocol. Participants were standing / sitting? Height of tubing was adjustable?

l. 115: tests were performed in the morning or during the entire day?

l. 150: QQ-plots?

l. 155: Pls, explain meaning of 0.8. Is there a graph?

Reviewer #2: The authors did a great job at comparing different methods of determining CFF thresholds, considering individual and group variability. Given the relatively widespread use of the flicker test and the lack of unified paradigms between studies, I think the manuscript is an interesting contribution to the current state of knowledge. The authors rightly pointed out the problem associated with the long time duration of the method of constant stimuli, which would be difficult to apply in some conditions, such as hyperbaric conditions.

The description of the methods is sound. Taking repeated measurements at a similar time of day for each participant is clearly an advantage of the study. The manuscript may benefit from adding of more precise information about the hours of the study.

Statistical analyses are presented in a sound manner, but I suggest adding a brief note about the "car package" from R (line 148/149), explaining that it is a set of functions and tools for regression analysis.

I also recommend moving the description of normality in the results section to the very beginning, before the description of correlations to improve the readability of the description and justify the use of parametric tests.

The created graphs are readable, but I think the data presented in Figure 1 would be more easily compared visually after using the same reference points on the x and y axes.

Description of Figure 3 - please standardize the lowercase and uppercase letters in the statistics given.

In the discussion section, the authors describe factors that may affect the differences in CFF thresholds. I agree that they may be related to neuroanatomical, neurophysiological and neuropsychological factors. Have previous studies compared genders or age groups in this regard? I lack such an explanation.

It is not clear what the authors meant when they wrote about testing visual temporal resolution among humans in real-world conditions (line 292-294). I suggest adding a clarification.

In summary, I think the manuscript will be suitable for publication after making minor revision.

6. PLOS authors have the option to publish the peer review history of their article (what does this mean?). If published, this will include your full peer review and any attached files.

Reviewer #1: **Yes: **Prof. Dr. Jochen D Schipke

Reviewer #2: **Yes: **Natalia Daria Mankowska

---

## [Author Response · Author response to Decision Letter 0]

12 Jan 2024

Dear editor and reviewers,

Thank you for reviewing our submission and for offering comments and suggestions for improvement of the manuscript. We have made all the suggested changes and have addressed each comment in the uploaded document titled 'Response to Reviewers.'

---

## [Editor Report · Decision Letter 1]

17 Jan 2024

The speed of sight: individual variation in critical flicker fusion thresholds

PONE-D-23-39116R1

Dear Dr. Haarlem,

We’re pleased to inform you that your manuscript has been judged scientifically suitable for publication and will be formally accepted for publication once it meets all outstanding technical requirements.

Kind regards,

Christopher Nice, Ph.D.

Academic Editor

PLOS ONE
---

## [Editor Report · Acceptance letter]

22 Mar 2024

PONE-D-23-39116R1 

PLOS ONE

Dear Dr. Haarlem, 

I'm pleased to inform you that your manuscript has been deemed suitable for publication in PLOS ONE. Congratulations! Your manuscript is now being handed over to our production team.

Kind regards, 

on behalf of

Dr. Christopher Nice 

Academic Editor

PLOS ONE